# SMEs Awareness and Preparation for Digital Transformation: Exploring Business Opportunities for Entrepreneurs in Saudi Arabia's Ha'il Region

**Abhishek Tripathi** * and **Ajay Singh** *

Department of Management & Information Systems, College of Business Administration, University of Ha'il, Hail 55476, Saudi Arabia
* Correspondence: a.tripathi@uoh.edu.sa (A.T.); a.singh@uoh.edu.sa (A.S.)

**Abstract:** This study investigates how Saudi SMEs' awareness and preparation for digital transformation impact their business performance (BP). First and foremost, in this study, we examine the impact of awareness and preparation on business performance using the intention to use (ITU) as an initial mediator. In addition, the researchers also examined the impact of awareness and preparation on business performance through intention to use and Government Support (GS), and later, intention to use and skills as serial mediators. Using a structured questionnaire based on a seven-point Likert scale, data were collected from 68 SMEs in the Ha'il region of Saudi Arabia, one of the Kingdom's leading economic provinces. The data were analyzed through simple and serial mediation techniques through AMOS-24. The study found that SMEs' awareness and preparation for digital transformation significantly and positively influenced their intentions to use the technology. The researchers found that intention to use, skills, and government support are significant variables that improve business performance. The research also revealed full-serial mediation between awareness and business performance and preparation and business performance, showing that intention to use and skills and intention to use and government support significantly mediate improving SMEs' business performance. The study implications provide for SMEs' successful digital transformation, considering the role of skills and government support, which will help SMEs improve their performance and embrace sustainability in human and economic development in Saudi Arabia. Together with policymakers, SMEs, and researchers, it will also look at the entrepreneurial potential for Saudi nationals in the run-up to Vision 2030.

**Keywords:** digitization; SMEs; digital tools; intention to use; business performance; skills; government support; sustainability

## 1. Introduction

Digitalization is perceived as an essential technological strategy that will drastically transform the industry by substantially improving the entire value chain. SMEs, however, require more clarity about digitalization complexity and costs, so the implementation process needs to be streamlined. Baiyere and Hukal (2020) [1] stated, "Digitalization transforms a business model and creates new opportunities to generate value and revenue". An important component of digital entrepreneurship is the transference of assets, services, and significant parts of a business to a digital format [2]. Emerging economies rely on small- and medium-sized businesses to drive sustainable economic growth, create jobs [3], and increasingly contribute to the developing nations' gross domestic product (GDP) [4]. The non-oil GDP of the Kingdom is primarily composed of SMEs, accounting for about 21% of GDP. This is less than the average of 46% among the top 15% of worldwide economies [5]. Digital technology has been demonstrated to play a vital role in digital transformation strategy and organizational innovation [6]. The concept of digital entrepreneurship emerged from technologies such as communications and information technology. As a result of the

internet, entrepreneurship is now possible worldwide, resulting in greater competition. Digitalization of small business enterprises and their overall processes, including artificial intelligence (AI), lie across both public and private sectors and is vital in transforming Saudi Arabia's Vision 2030 [7]. Unlike traditional entrepreneurship, entrepreneurs invest in digital competition because it does not require vast capital [8]. In addition to Saudi Arabia and other Gulf consortium countries, which possess rich natural resources but are susceptible to commodity price fluctuations, the diversification of the economy of SMEs at domestic, regional, and international levels will result in long-term financial benefits [9].

Digitalization and sustainability are two key focus areas for the industrial sector that are essential to long-term commercial success because digitization creates economic sustainability [10]. Corporate economic sustainability transformation has become much more relevant recently, even though digital transformation has been a major item on the strategic agenda of many firms for a number of years [11]. Technology advancements in infrastructure create several opportunities for entrepreneurs interested in digital entrepreneurship. However, digital entrepreneurship opportunities, barriers, and success factors regarding new digital enterprise models are receiving little attention from society [12]. Recent socioeconomic developments, such as network effects, will make it increasingly important for businesses to identify opportunities in the digital world [13]. The association of companies and entrepreneurs' digitalization has resulted from increasing business activities over time [14]. The provision has been made to increase the contribution of SMEs in the Kingdom following its National Transform Program [15]. Recognizing and leveraging digital technology (DT) opportunities is a competitive necessity in the digital world. However, the influence of digital technology makes it difficult to identify opportunities due to agency dispersions and the blurring of boundaries among customers and industries [16], suggesting that big-data analysis and artificial intelligence will form the basis of future digital entrepreneurship.

The macro trends of sustainability and digitization influence today's economy and society and necessitate significant changes [17,18]. The digital era, which is transforming society and the economy, has arrived due to technology's rapid and ongoing advancement [19]. Gregori and Holzmann (2019) [20] contend that digitalization implies two things: first, a developing digital logic that is distinct from sustainability logic; and second, a supporting logic for the creation of value propositions that incorporate the three main components of sustainable development: social, environmental, and economic value. It is important to consider entrepreneurship as a digital, social, and economic change tool. Entrepreneurial mindsets can benefit the digital and environmental transitions [21]. Increased knowledge of opportunities and entrepreneurial skills while acknowledging the complex ecosystem in which they are embedded might incite organizations to begin their digital transformation [22–24]. Digitalization has positive sociocultural effects, including increased inclusion in rural communities, primarily due to connectedness and improved well-being, and a smart community oversees cutting-edge infrastructure and technologies, facilitating the development of interpersonal dynamics that extend from the individual to the community as a whole [25]. Digitalization in the manufacturing sector can positively impact sustainability development, provided that the problems posed by technology and societal developments are effectively tackled, as acknowledged by Chen and Kamal [26] and Lee et al. [27]. Digitalization fundamentally alters social ideas of the job market [28]; MSMEs (Micro-, Small-, and Medium-sized Enterprises) with digital skills have access to the resources and tools they need to take advantage of digital technology, streamline operations, and promote sustainable practices. UN Sustainable Development Goal 9 calls for creating a resilient infrastructure, expanding Internet connectivity to the least-developed nations, advancing technological capabilities, and encouraging small-scale businesses and industries to participate in global value chains [29].

Sustainability and digitalization are major concerns for investors and businesses, and are becoming increasingly intertwined. Digital technology has a key role in the development of companies and in the management of their environmental footprints. By offering

real-time operational data and information on the effects of a business's processes and activities, digitization generally aids in sustainability management. Data analytics, artificial intelligence (AI), and simulation technologies can be used to handle and analyze this data and information. Examples of these technologies include the Internet of Things (IoT) and traditional enterprise resource planning systems [30]. From a managerial perspective, business owners can increase productivity by optimizing their processes and cutting labor costs by substituting automated tasks for manual ones. Technological infrastructures enable economies of scale that might be advantageous to business owners as well [25]. The convergence of digitalization with sustainability presents novel prospects for tackling worldwide issues, establishing a fair and enduring community, and establishing the foundation for accomplishing the Sustainable Development Goals (SDGs) [31]. The digital transformation that promotes sustainability is crucial for modern companies. By adopting sustainable practices, MSMEs can enhance their competitive edge by drawing clients and investors with similar beliefs. It also allows MSMEs to innovate, cut expenses, and improve their brand using sustainable practices [32].

In addition to providing more access to education and culture and lowering regional inequities, the digital revolution offers enormous potential for increased productivity, innovation, and employment. It also presents significant social and environmental opportunities [33]. The digital marketplace makes efficient transactions possible, integrating economic and socio-environmental value by guaranteeing producers profit even when they give fair prices [20]. The Organization for Economic Co-operation and Development's (OECD) rural development policy highlights the relevance of digitization for sustainable development, highlighting its critical role in rural realities. The government of Saudi Arabia is concerned with developing its local talent to provide its youth with job opportunities in innovative activities, ensuring human and economic sustainability, leading to Saudi Vision 2030 [34]. Industry 4.0-related digital tools like artificial intelligence (AI), big data and analytics, cloud computing, etc., are being used to promote the growth of the digital economy. This leads to better consumer experiences, increased company performance for SMEs, and helps the country create economic sustainability [35]. The Saudi government's digitalization initiatives aimed to enhance human capital, or social skills and competency, in order to facilitate digitization [10,36].

The research emphasizes SMEs' growth and performance with the Kingdom's current business opportunities and challenges. In this regard, the digitalization of SMEs becomes an immediate challenge for the country in association with its Vision 2030, adopting industry 4.0 and 5.0 aspects [37,38] over the prevailing traditional SME performance perspectives. Based on the available literature, very few studies have assessed SMEs' performance in Saudi Arabia within the context of digital transformation awareness and preparation. The study's novelty is that it investigates the SMEs' awareness [39] and preparation to transform their performance by examining the role of intention to use, government policies, and SMEs' skills. The study shows its uniqueness in measuring SMEs' performance by filling the gap by examining the mediation effect, adding value to the existing literature, and contributing to the body of knowledge.

The study structure consists of an Introduction, Literature review, Theoretical perspective, Research Questions, Hypothesis, Research Methodology, Results, Discussion, Practical implications, Conclusion, Limitations, and Prospects for Future Research.

## 2. Literature Review

The widespread acceptance of SMEs' digital transformation is expected to vastly transform the industry by substantially refining its value chain [40]. To maximize business competitiveness and create growth opportunities through innovation [41], businesses can leverage emerging digital technologies based on a mixture of market, learning, and entrepreneurial orientations. Digitization is the most critical force in entrepreneurship and innovation, and it can be the most crucial factor in thriving; likewise, so is the COVID era [42]. A digitally networked society requires digital entrepreneurs to develop their

entrepreneurial capacity through social networks and investigate the relationship between digital technology and entrepreneurship support policies, emphasizing the role of interaction between the two [43,44]. The study by Abebe (2014) [45] provides empirical evidence on the roles played by e-commerce adoption and entrepreneurial orientation within small firms. Keeping up with the fast-paced development of online technology can help improve brand recognition, boost business profiles, and open new avenues for revenue. People join a social network to discuss a subject of mutual interest on a social media site [46]; engaging critical consumers and creating brand advocates can be a powerful method for attracting influential consumers.

Irajifar et al. (2023) [31] examined a strong relationship between digitalization and sustainability. The study revealed four important digitization pillars that contribute to sustainable economic growth. These pillars are governance, energy, innovation, and systems. The study found that government support, or government role, plays a significant role in improving SMEs' business performance. Digitization helps enhance production, lowers the cost of production, and promotes green globalization, which helps enhance sustainable development [35]. According to Brenner and Hartl (2021) [17], digitalization and sustainability are macro trends reshaping society and the economy and necessitating significant changes. Using bibliometric analysis in the Scopus database, Lertpiromsuk and Ueasangkomsate (2022) [47] classified regions for crucial terms in three areas: sustainable digital economy, sustainable manufacturing development, and sustainable business innovation.

## 2.1. SMEs Awareness and Preparation

The SMEs' awareness and spirit of entrepreneurship are essential for sustainable economic growth and development; therefore, receiving more professional assistance and raising awareness could be crucial to their success [39]. Azevedo and Almeida (2021) [48] investigated that the major obstacle is a need for more awareness regarding the potential and implications of digital technologies among decision-makers. Lukonga (2020) [49] found that many SMEs are unfamiliar with digitalization and that terms, concepts, and theories are often challenging to comprehend. However, Kergroach's (2020) [50] study found that SMEs lack information and awareness about digital transformation and skills to identify and manage change within their businesses. Additionally, the study stated that some firms know different terms and concepts related to digitalization but are unaware of how to put them into practice, showing that they have theoretical knowledge but not practical knowledge. On the other hand, due to the Coronavirus disease 2019 (COVID-19), small- and medium-sized enterprises have accelerated digital transformations [51]. The disruption of supply chains during lockdowns and social distancing forced businesses to rethink business models and move operations online or implement innovative working solutions [52]. A study by Skare et al. (2023) [53] shows that SMEs' awareness of digital transformation improves business performance, overcomes challenges, and reaches mass markets. However, a study conducted by Garzoni et al. (2020) [54] in south Italy among 1000 SMEs found that they are not fully aware of the benefits of adopting digital transformation, which is a major concern. There is an urgent need to increase awareness of digital tools, which help them to improve their productivity or efficiency. A four-phase digital transformation model by Szopa and Cyplik (2020) [55] assists SMEs in preparing for smooth technology implementation, improving their supply chain. Hong et al. (2012) [56] emphasized how SMEs react, plan, recover, and expand during times of crisis.

## 2.2. Digitalization and Intention to Use

A study by Teng et al. (2022) [57] found that employees' digital skills, technology implementation, and digital transformation strategy development are three crucial resources for SMEs' growth and development. A study conducted by OECD (2021) [52] on "The Digital Transformation of SME" indicates that SMEs face mistrust and uncertainty about these technologies (i.e., the operational risks of SMEs can be unforeseen, or they can be

locked into online platforms), like sudden changes in platform policies and server outages. It may also be difficult for businesses to understand customer data online, as they do not have direct access to customer data [58]. Also, the different fees charged by different online platforms may affect the profitability of SMEs and harm their competitive position compared to larger companies with more bargaining power to negotiate for lower fees [52]. SME supply chains will likely be deeply affected as SMEs implement digitalization, increasing their exposure to online attacks. SMEs lack access to external consultants due to financial constraints [59] and specialized IT professionals capable of maximizing digital transformation tools [60]. Mertens and Thiemann (2019) [61] emphasize that SMEs must learn about available solutions and their potential advantages. Maduku et al. (2016) [62] found that adopting new technologies among SMEs is more likely when they see an opportunity to increase the efficiency of their entire business operations. A changing and competitive environment has required SMEs to modify their business skills to compete in regional and global markets [36,63]. In addition to transferring products and buying, selling, and exchanging information, SMEs intend to adopt and use digitization [64]. Previous research indicates that individuals implementing improved management strategies for technology adoption and business performance are more likely to integrate digital and green technologies following "environmental, social, and governance (ESG)" metrics to monitor corporate governance using these emerging platforms [65]. According to Gregori and Holzmann, "value spillover offers new perspectives on entrepreneurial value creation for sustainability alongside the role of digital technologies in enabling the formation of societies, co-creation activities, and stakeholders' integration" [20]. Digital technologies are used in many different areas, including banking, peer-to-peer services (economy), food production, power, housing, healthcare, and mobility, and they are an essential element of peoples' lives and the life of organizations and institutions [66].

### 2.3. Skills and Government Support

Digital skills aim to identify, access, and create new knowledge using digital tools and technologies [8,67,68]. Due to the lack of funding, skills, and human resources, SMEs underutilize digital technologies and fail to adopt them [69–72]. Lukonga (2020) [49] recommended that the large and growing digital skill gaps represent another significant barrier to SMEs adopting digital solutions. Monsha'at [73] reported a gap in cloud computing, artificial intelligence, machine learning, mobile tech, blockchain tech, data analytics, and advanced security. Telukdarie et al. (2023) [74] investigated that SMEs know digitization and its benefits, but need more skills, time, and finances. SME growth and innovation performance are influenced by personal digital ability [75]. The Global Statistics (2024) [76] reported that 35.97 million people in Saudi Arabia have Internet access, representing 99% of the total population. According to Kemp (2023) [77], 29.10 million Kingdom of Saudi Arabia (KSA) social media users were active in January 2023. Instagram has 27.88 million active users in Saudi Arabia, while WhatsApp has 30.39 million active users [76]. Therefore, to increase revenue, digitalization is recommended [78]. Singh and Alshammari (2023) [79], in his study, considered government policies as a mediating variable between e-technology and social empowerment, between e-technology and reshaping Saudi society, and between e-technology and advancement of Saudi society, and found that the government initiatives have been playing a positive significant role in the social empowerment, advancement, and transformation of the national human resources complying with Saudi vision 2023. According to Vial (2019) [80], digital transformation (DT) is a disruptive process that occurs when organizations adopt digital technologies to change value-creating processes in response to changes in the business environment. DT encourages innovation because it necessitates the acquiring new knowledge and skills, fosters new collaboration within the organization, and supports creating new business models [81]. Digital transformation is a process that results in new institutional arrangements, values, practices, and structures. Typical examples of this include standard-setting digital infrastructures like product platforms and blockchain technology or widely accepted and configurable digital modules

like ERP systems [82]. With knowledge, leadership, digital servitization, and technological aspects, digital transformation can potentially facilitate internationalization [83].

The role of SMEs in long-term, broadly shared economic development is increasingly recognized by governments worldwide [84]. Governments have created regulatory and institutional frameworks that are pro-competition for SMEs [52]. Chang et al. (2017) [85] suggested that the government could implement policies and programs aimed at helping small service businesses transform to digital technologies, as well as how to spend the government's annual budget to assist small businesses in adopting digital technologies. Saudi Arabia authorities are working hard to build resilient infrastructure, promoting innovation and sustainable industrialization (SDG-9) and providing a plethora of opportunities for SMEs to grow nationally or globally. The industrial development and digital transformation are significantly shaped by national policies [86]. Improved business performance leads to sustainable economic development in the country. While sustainability-related macroenvironmental features consistently impact the corporate agenda, SMEs are indirectly forced to implement digital tools committed to implementing a digitization plan, which indirectly relates to economic sustainability [87].

### 2.4. SME Business Performance

Research conducted by Yu et al. (2022) [88] explores that digital production technology's real-world effect of enabling enterprises to connect with their stakeholders in a diverse and personalized way in the shortest possible time can be explained using advanced integrated platforms. Due to the fourth industrial revolution, organizations can improve their product and service quality, improve their processes, lower costs, easily modify products, and attract more customers according to their preferences and needs [89]. A key benefit of digitization for SMEs is that it gives them greater access to innovation and allows them to generate and analyze data in new ways to enhance their performance [52]. Helfat and Raubitschek (2018) [90] found that digital tools could enhance customer interaction and improve product-service systems. Through digital technology, new digital products and services can be developed, resulting in a broader client base and improved performance of SMEs [91]. In light of the rise of the digital economy, small companies can now participate more actively in global value chains [92]. Curraj (2017) [93] found that digitalization and SME performance have positive and significant relations. Through the digital operation and integration of various technologies, digitally enabled organizations would improve their business transformation efforts [94,95].

The factors are displayed in Table 1, along with the references and descriptions used in the study. Study components include awareness, preparation, intention to use, skills, and government support. The study focuses on how SMEs' awareness and preparation towards digitalization impact their intention to use. Furthermore, the study explores the impact of "intention to use" on their business performance. The researchers tried to establish a link between SME awareness and/or preparation for digitalization and business performance with the mediating role of government support and skills provided to SMEs in Saudi Arabia. Finally, the relevant strategies are suggested to boost the digitization of SMEs in the Kingdom. From the above Table, several variables from the constructs, i.e., (Awareness 2 and 5; Preparation 3 and 4; Intention to use 1 and 4; Business Performance 3 and 5; Skills 1 and 3; Government support 1) have been removed since the factor loadings are low (<0.50).

**Table 1.** Exploring Variables of the Model.

| Study Variables | References | Description | Statements |
|---|---|---|---|
| SME awareness and Preparation | [36,39,48–56] | Awareness regarding digital technologies; awareness of digital terms; concepts and theories; awareness and professional assistance, lack of information and awareness and skills about digital transformation; economic sustainability and development; preparation for digital transformation. SMEs preparation for crisis management. Sustainable economic growth and expansion. | • **Awareness_1:** Have you ever attended any course, lecture, or training to increase awareness of digitization?<br>• **Awareness _2:** Our organization deploy applications on cloud infrastructure.<br>• **Awareness 3:** I am aware of the available tools for digitization.<br>• **Awareness_4:** We know the new digital possibilities and can identify the right technology options for business growth.<br>• **Awareness_5:** We are aware of the strategic online marketing models for expanding the business activities of SMEs digitally?<br>• **Preparation_1:** We are making relevant preparations to implement digitization that will help us gain a competitive advantage.<br>• **Preparation_2:** We are improving our base so that SMEs can compete on closer terms with larger organizations.<br>• **Preparation_3:** Due to digitization, employee productivity will improve.<br>• **Preparation_4:** Digitization is an opportunity to protect itself from threatening competitors.<br>• **Preparation_5:** We are preparing for Digitization because it will help small and medium-sized enterprises (SMEs) integrate into global markets. |
| Digitalization and Intention to Use | [20,31,35,40–42,44–46,52,57–62,64–66,96,97] | Personal digital ability; digital innovation; knowledge; internationalization; cost reduction; improved production capacity; increased workforce productivity and output; visionary and customer-oriented; business survival and ecological; adaptability to ever-changing social norms, technological advancements, and product market conditions; improved brand awareness, customer loyalty; revenue generation; value creation. | • **Intention_1:** We will continue to invest in digital projects.<br>• **Intention_2:** We will recommend others to invest in digital projects.<br>• **Intention_3:** We intend to use digital tools regularly.<br>• **Intention_4:** Access information quickly and enhance communication networks across the globe.<br>• **Intention_5:** I plan to implement digitization in the future |

**Table 1.** *Cont.*

| Study Variables | References | Description | Statements |
|---|---|---|---|
| Skills and Government Support | [8,49,52,67–87] | developing unique skills; improve the market position; Company's growth, uncertainty avoidance; digital operation and integration; growth and innovation. long-term economic development; special programs; funding; policies and counselling; training programs; development initiatives; government support. | • **Skill_1:** I have digital skills and knowledge to implement digitization<br>• **Skill_2:** We have skills to accelerate digitization and enable us to be technically sound to build new digital solutions.<br>• **Skill_3:** The SMEs expect to control information, make decisions, and tell their subordinates what to do.<br>• **Skill_4:** We have managerial skills for the effective implementation of digitization.<br>• **Skill_5:** We have the financial skills for the effective implementation of digitization.<br>• **GS_1:** The government organizes competitions that reward and motivate traditional SMEs using digital tools.<br>• **GS_2:** The government provides various benefits to implement digitization.<br>• **GS_3:** There are various mobile applications to integrate small enterprises into the digital market.<br>• **GS_4:** The government has established entrepreneurial platforms (such as business incubators & accelerators) dedicated to the digital transformation of SMEs. |
| SME Business Performance | [88–95] | Connect with the stakeholders; greater access to innovation; new digital products and services; positive relations; enhance productivity, innovation; access to international markets; operational efficiency; cost reduction; improved production capacity; increased output of the workforce; value creation; developing unique skills, improved the market position; Company's growth; digital operation and integration; competitive edge. | • **BP_1:** Digitization helps build relationships and enhance logistical integration.<br>• **BP_2:** Digitization helps communicate to a wider range of customers for better productivity.<br>• **BP_3:** Digital platforms provide an excellent ability to grow a SMEs brand.<br>• **BP_4:** Digital technology will improve overall business performance, including customer experience.<br>• **BP_5:** The implementation of digitization helps in improving return on investment. |

## 2.5. Rationale of the Study

A thorough literature review found that most studies have been conducted in Western countries. Despite this, a few studies and reports focus on the Saudi SMEs' towards digitization [5,7,15,34,36,49,52,73,75–77,94,98,99]. The study focuses on the Ha'il province, located in the northern region of Saudi Arabia, having 103,887 km$^2$ estimated area, and 746,406 population [100]. It is one of the major economic cities in the Kingdom of Saudi Arabia. The study also emphasizes the role of skills and government support in successfully implementing digitization to improve SMEs' business performance. The researchers integrated several models relating to awareness and preparation for digital transformation to assist SMEs in improving their business performance and assist the authorities in successfully implementing the national transformation program. While previous studies and reports attempted to elaborate on digitalization about business performance in Saudi Arabia, the present study shows its uniqueness in measuring SMEs' performance by filling the gap by examining the mediation effect of government and skills with SMEs' perception

and preparation and intention to use (digitalization), adding value to the existing literature and contributing to the body of knowledge.

## 3. Theoretical Perspectives

To measure the performance of small and medium-sized businesses (SMEs), the study emphasizes adopting new technology (digitalization) through the application of the theory of reasoned action (TRA) [101], the Technological Acceptance Model (TAM) [102], and theory of planned behavior (TPB) [103].

In the research, the Theory of Reasoned Action (TRA) is primarily applied to explain awareness, attitude, intention, and behavior, i.e., (awareness- attitude- intention- behavior) [104]. The TRA theory is deterministic awareness of the SMEs about digitalization (awareness) and preparation of SMEs towards digitalization (attitude) in order to measure the performance of SMEs (Behavior). Secondly, the research applies the Technological acceptance model (TAM), which is deterministic, given by Davis (1989) [102], a widely used theory based on information and communication technology (ICT) that determines the adoption of IT base behavior. The study emphasizes the preparation (perceived ease of use in the adoption of the digital technology for SMEs) and awareness (perceived usefulness of the SMEs' digitalization) leading to SMEs' digitalization (Behavioral intent to use), resulting in the Performance of SMEs (behavior), Thirdly, the theory of planned behavior (TPB) predicts planned behavior related to decision-making (the SMEs' performance). The digitalization of SMEs intends to perform the expected behavior of digital transformation in terms of valuable outcomes (SMEs' performance). The preparation and awareness of SMEs relate to behavior (attitude, normative beliefs towards digitalization) and perceived behavior (control beliefs), along with the subjective norms (barriers and control beliefs), which result in the perception of individual behavior to perform/omit the particular behavior (digitalization). The research emphasizes the awareness and preparation of SMEs in adopting digitalization (technology and people) towards the societal and industrial change associated with that emerging perspective, leading to the performance of SMEs based on the principle of process, which shapes the outcome with the human-technology interaction. This theory is applied to view the role of digitalization in shaping the SMEs, leading to social and industrial empowerment.

## 4. Research Objectives

1. To study the awareness and preparation of SMEs toward digital transformation in the Saudi Arabia.
2. To explore the opportunities for entrepreneurs and their impact on SMEs' performance and digitalization.
3. To suggest strategies for the growth of SMEs by implementing digital marketing tools in the Kingdom.

Based on the above research objectives, the study derived the following research questions.

*Research Questions*

1. How do SMEs' awareness and preparation about digitalization impact their intention to use?
2. What impact does digitalization (intention to use) have on the business performance of a small or medium-sized business?
3. Is there a link between SME awareness and/or preparation, and business performance, and the government support and skills provided to SMEs?
4. What strategies should the government pursue to boost the digitization of SMEs in Saudi Arabia?

## 5. Hypothesis

**H₁:** *SMEs' awareness towards digital transformation significantly impacts intention to use.*

**H₂:** *SMEs' preparation for digital transformation significantly impacts their intention to use.*

**H₃:** *SMEs' Intention to use digital transformation impacts business performance.*

**H₄:** *Intention to use mediates the relationship between awareness and business Performance.*

**H₅:** *Intention to use mediates the relationship between preparation and business performance.*

**H₆:** *Intention to use and skills fully mediate the relationship between SMEs' awareness and business performance.*

**H₇:** *Intention to use and government support fully mediate the relationship between SMEs' awareness and business performance.*

**H₈:** *Intention to use and Skills fully mediate the relationship between SMEs' preparation and business performance.*

**H₉:** *Intention to use and government support fully mediate the relationship between SMEs' preparation and business performance.*

## 6. Research Methodology

Initially, the study develops a conceptual model based on the literature review to determine the proposed outcome to justify the research to fill the gap. The research applied a survey method to collect the data of SMEs to get a suitable outcome. It uses SPSS AMOS 24 quantitative software to analyze the data. Finally, the study concludes and presents the outcome accordingly.

### 6.1. Pilot Survey

The survey questionnaire was initially distributed among three experts consisting of entrepreneurs (two) and management professors (one) to justify the questions and structure of the questionnaire suitable for the survey to get the robust outcome of the research on SMEs' performance concerning awareness and preparation about SMEs' digitalization in the Ha'il region of Kingdom of Saudi Arabia (KSA). Subsequently, the researchers modified the questionnaire based on the experts' proposed recommendations for the survey from the respondents.

### 6.2. Sampling and Data Collection

The survey population consists of owners from various SMEs in the Ha'il region of Saudi Arabia using Judgmental non-probability sampling. The reason for using judgmental sampling is that the owners can better give insight and respond appropriately. The researcher contacted 112 SMEs in the Ha'il region for the online survey, of which 68 SME owners, i.e., 60.71%, agreed and responded through the online Google form survey questionnaire. The study used a 22-item questionnaire with a 7-point Likert scale.

### 6.3. Data Analysis

The study uses SPSS AMOS 24 software for the quantitative data analysis collected from the respective SMEs' respondents. Further, the study incorporated the interpretation of the results in the discussion part. The study applied three mediators: intention to use, government policies, and skills. The study presents results, discussions, conclusions, limitations, and practical implications.

### 6.4. Measurement Model

Figure 1 illustrates the study measurement model focusing on how SMEs' awareness and preparation improve business performance. The study uses intention to use, govern-

ment support, and skills as mediators to determine whether these variables are essential in implementing digitalization and improving SME performance in the Ha'il region.

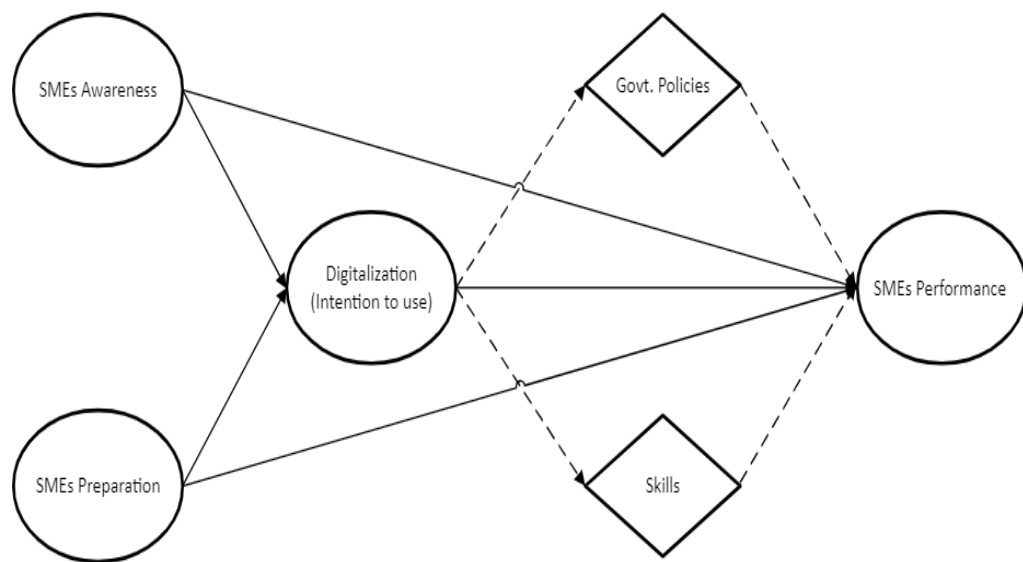

**Figure 1.** Measurement Model.

## 7. Results

Table 2 shows the entrepreneur's demographic profile. The data were collected from 68 SME owners. The study found that 76.47% of the participants were males and 23.53% were females; 2.9% of respondents had a high school diploma, 41.2% had a diploma, 48.5% had a bachelor's degree, and 7.4% had a master's degree. About 38% of respondents have been in entrepreneurship for at least five years; 35.3% of SMEs have four to five years of experience; 5.9% have two to three years; and 2.9% have one year.

**Table 2.** SMEs entrepreneur demographic profile.

|  |  | Frequency | Percent |
|---|---|---|---|
| Gender | Male | 52 | 76.47% |
|  | Female | 16 | 23.53% |
| Qualification | High school | 2 | 2.9% |
|  | Diploma | 28 | 41.2% |
|  | Bachelors | 33 | 48.5% |
|  | Master's degree | 5 | 7.4% |
| Age | 26–35 year | 5 | 7.4% |
|  | 36–45 year | 26 | 38.2% |
|  | Above 45 years | 37 | 54.4% |
| Experience in entrepreneurship | One year | 2 | 2.9% |
|  | 2–3 years | 4 | 5.9% |
|  | 4–5 years | 24 | 35.3% |
|  | Five years and above | 38 | 38% |

The researchers used AMOS to compute Confirmatory Factor Analysis (CFA). Several variables from the constructs (Awareness 2 and 5; Preparation 3 and 4; Intention to use 1 and 4; Business Performance 3 and 5; Skills 1 and 3; Government support 1) have been removed since the factor loadings are low (<0.50). Model fit measures (CMIN/df, CFI, GFI, TLI, SRMR, and RMSEA) were used to determine the model's overall goodness of fit. All values were within the expected acceptance levels, i.e., ($\geq$0.50) [105–107]. The model yielded a good fit.

### 7.1. Construct Reliability

Table 3 shows the results of factor loadings, reliability, and average variance extracted from SPSS Amos 24. Cronbach's Alpha and Composite Reliability were used to assess construct reliability. All constructs in the study had Cronbach Alphas exceeding the limit of 0.70 [108]. The threshold value of composite reliability ranged from 0.746 to 0.794, and our values exceeded the threshold value (>0.70) [109]. Therefore, each construct was found to be reliable.

**Table 3.** Loadings, reliability, and convergent validity.

| Constructs and Their Items | Factor Loading | Cronbach Alpha | CR | AVE |
|---|---|---|---|---|
| Awareness | | | | |
| • Awareness_1 | 0.772 | | | |
| • Awareness_3 | 0.632 | 0.769 | 0.770 | 0.529 |
| • Awareness_4 | 0.770 | | | |
| Preparation | | | | |
| • Preparation_1 | 0.611 | | | |
| • Preparation_2 | 0.844 | 0.736 | 0.756 | 0.513 |
| • Preparation_5 | 0.673 | | | |
| Business Performance (BP) | | | | |
| • BP_1 | 0.781 | | | |
| • BP_2 | 0.715 | 0.757 | 0.794 | 0.563 |
| • BP_4 | 0.754 | | | |
| Intention to use (ITU) | | | | |
| • ITU_2 | 0.681 | | | |
| • ITU_3 | 0.727 | 0.753 | 0.754 | 0.506 |
| • ITU_5 | 0.708 | | | |
| Skills | | | | |
| • Skill_2 | 0.849 | | | |
| • Skill_4 | 0.539 | 0.836 | 0.746 | 0.502 |
| • Skill_5 | 0.704 | | | |
| Government Support (GS) | | | | |
| • GS_2 | 0.761 | | | |
| • GS_3 | 0.792 | 0.876 | 0.785 | 0.551 |
| • GS_4 | 0.668 | | | |

Source: Values extracted from SPSS AMOS 24.

In order to estimate the convergent validity of scale items, the study applied the Fornell and Larcker (1981) [110] criteria. As a result of variance extraction (AVE), the average values exceeded the threshold value (≥0.50) [110]. Hence, the scales used in this study possess the required convergent validity.

### 7.2. Discriminant Validity

The notion of discriminant validity is argued by Hair Jr. et al. 2016 [111] by stating that it ensures a construct's uniqueness. Similarly, Hair et al. 2019 [112] indicated that discriminant validity is established when one construct has a greater shared variance than all the others. In order to satisfy this criterion, each construct's square root of AVE should be greater than the other constructs' highest correlations [111,112]. As a result, this study establishes discriminant validity by using values for each construct represented in Table 4 [110].

**Table 4.** Discriminant validity of constructs (Fornell–Larcker Criterion).

| | Awareness | Preparation | ITU | Skills | GS | BP |
|---|---|---|---|---|---|---|
| **Awareness** | 0.727 | | | | | |
| **Preparation** | 0.233 | 0.716 | | | | |
| **ITU** | 0.492 | 0.508 | 0.711 | | | |
| **Skills** | 0.141 | 0.082 | 0.130 | 0.709 | | |
| **GS** | 0.110 | 0.294 | 0.381 | 0.272 | 0.742 | |
| **BP** | 0.187 | 0.181 | 0.399 | 0.131 | 0.217 | 0.751 |

Source: Generated from SPSS Amos 24.

### 7.3. Structural Model

A structural model is defined by Hair Jr. et al. (2016) [111] as one that shows the relationship between constructs or latent variables in a study. Hair et al. (2019) [112] suggest that significant relationships must have t-values greater than 1.96.

Figure 2 illustrates the study's structural model. As a result of the model, we can see how awareness and preparation affect the intention to use and how it affects the performance of SMEs (BP). In Table 5, all fit indices are within the acceptable range: CMIN/df (Chi-square value) = 1.028 [107,113] GFI (goodness-of-fit index) = 0.900 [109] CFI (Confirmatory fit index) = 0.993 [105], TLI (Tucker and Lewis index) = 0.991 [105,114], SRMR (Standardized root mean square residual) = 0.068 [106], and RMSEA (Root Mean Square Error of Approximation) = 0.020 [106].

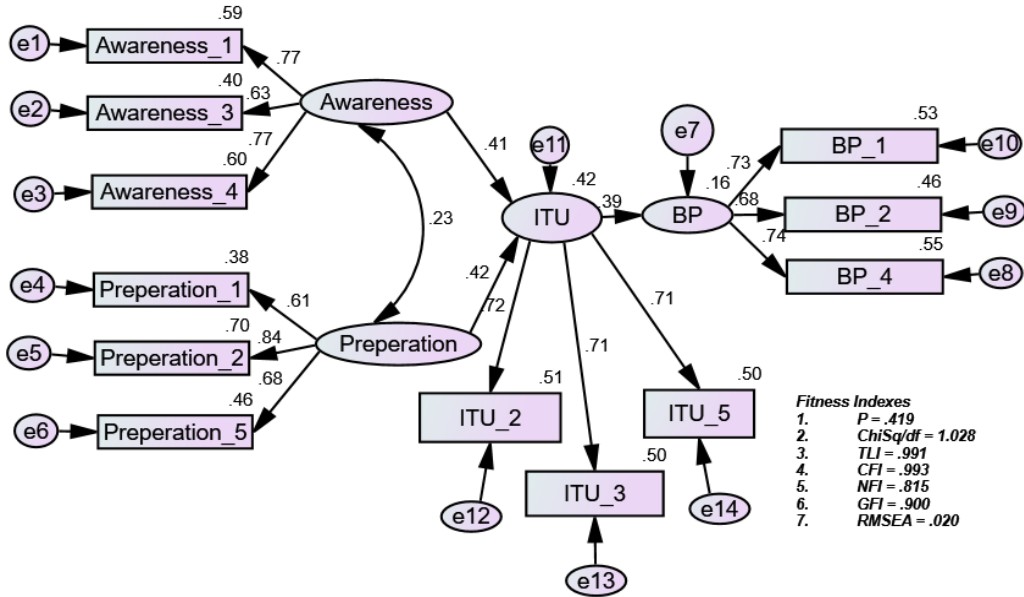

**Figure 2.** Structural model. Note: All fitness indexes achieved the required level. Source: Figure 2: Structural model generated through SPSS Amos 24.

**Table 5.** Model fit indices.

| Fit Indices | Good Fit If: | Sources | Obtained Value |
|:---:|:---:|:---:|:---:|
| P | *p* Value > 0.05 | Bagozzi & YI (1988) [115] | 0.419 |
| CMIN/df | 3–5 | Ullman (2001) [107], Less than 2 to 5 Schumacker & Lomax (2004) [113] | 1.028 |
| GFI | >0.90 | Hair et al. (2010) [109] | 0.900 |
| CFI | >0.90 | Bentler (1990) [105] | 0.993 |
| TLI | >0.90 | Bentler (1990) [105]; Tucker and Lewis (1973) [114] | 0.991 |
| SRMR | SRMR < 0.08 | Hu and Bentler (1998) [106] | 0.068 |
| RMSEA | RMSEA < 0.08 | Hu and Bentler (1998) [106] | 0.020 |
| | | Source: Generated from SPSS Amos 24 | |

Table 6 shows the model fit indexes and hypotheses in the study. The squared multiple correlations for intention to use were 0.424, indicating that awareness and preparation account for a 42.4% variance. The estimated value of business performance was 0.156, which shows that intention to use has accounted for a 15.6% variance in BP. This study evaluated the impact of awareness and preparation on intention to use and how it impacts business performance. The impact of awareness on intention to use was positive and significant (b = 0.412, t = 2.502, $p < 0.05$). Hence, $H_1$ is supported. The impact of preparation on intention to use was positive and significant (b = 0.418, t = 2.349, $p < 0.05$), supporting $H_2$. Intention to use significantly and positively impacted business performance (b = 0.395, t = 2.269, $p < 0.05$). Hence, $H_3$ is supported.

**Table 6.** Model fit indices and hypothesis.

| Hypothesized Relationship | Standardized Estimates (b) | *t*-Value | *p*-Value | Decision |
|---|---|---|---|---|
| Awareness -> ITU (Intention to Use) | 0.412 | 2.502 | 0.012 | Supported |
| Preparation -> ITU (Intention to Use) | 0.418 | 2.349 | 0.019 | Supported |
| ITU (Intention to Use) -> Business Performance | 0.395 | 2.269 | 0.023 | Supported |
| R-Square | | | | |
| ITU (Intention to Use) | 0.424 | | | |
| Business Performance (BP) | 0.156 | | | |

Model Fit: CMIN/df = 1.028, GFI = 0.900, CFI = 0.993, TLI = 0.991, SRMR = 0.068, and RMSEA = 0.020.

### 7.4. Mediation

In this study, we first identified the impact intention to use as a mediator to find the relationship between awareness and business performance as well as preparation and business performance.

Figure 3 and Table 7 represent the mediation analysis (Awareness -> ITU -> BP). The study assessed the mediating role of intention to use on awareness and BP. The result revealed an indirect effect of awareness and business performance with a positive and significant impact (b = 0.151, t = 2.086 *p* = 0.027), accepting H$_4$. We found insignificant direct effects on awareness and business performance in the presence of a mediator (b = 0.070, *p* = 0.717). This shows that the intention to use fully mediates the relationship between awareness and business performance.

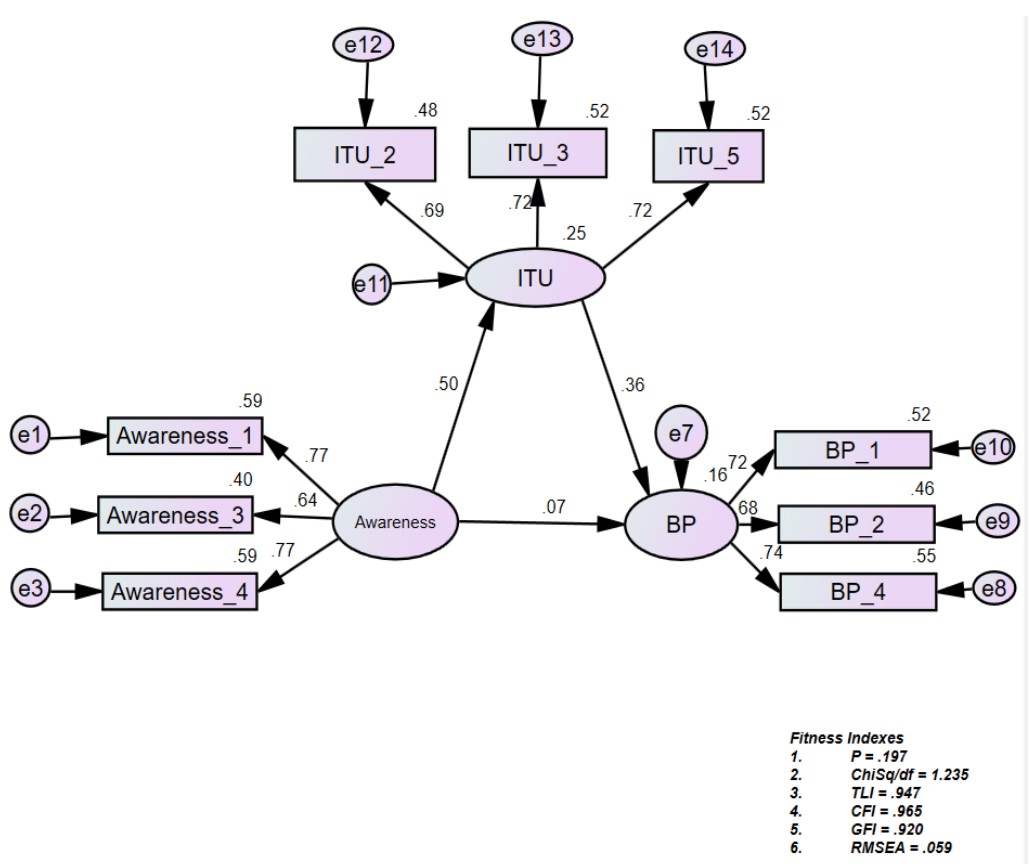

**Figure 3.** Representation of mediation analysis (Awareness -> ITU -> BP). Note: All fitness indexes achieved the required level. Source: Figure 3 generated through SPSS Amos 24.

**Table 7.** Mediating analysis summary (Awareness -> ITU -> BP).

| Relationship | Direct Effect | Indirect Effect | Confidence Interval | | *p*-Value | Conclusion |
| --- | --- | --- | --- | --- | --- | --- |
| | | | Lower Bound | Upper Bound | | |
| Awareness -> Intention to Use (ITU) -> Business Performance | 0.070 (0.717) | 0.151 | 0.023 | 0.384 | 0.027 | Full Mediation |
| | | Model Fit | | | | |
| CMIN/df = 1.235, GFI = 0.920, CFI = 0.965, TLI = 0.947, SRMR = 0.063 and RMSEA = 0.059. | | | | | | |

Figure 4 and Table 8 represent the mediation analysis (Preparation -> ITU -> BP). The study assessed the mediating role of intention to use on preparation and business performance. The result revealed an indirect effect of awareness and business performance with a positive and significant impact (b = 0.269, t = 2.23, *p* = 0.036), accepting $H_5$. We found positive but insignificant direct effects on awareness and business performance in the presence of a mediator (b = 0.041, *p* = 0.808). This shows that the intention to use fully mediates the relationship between preparation and business performance.

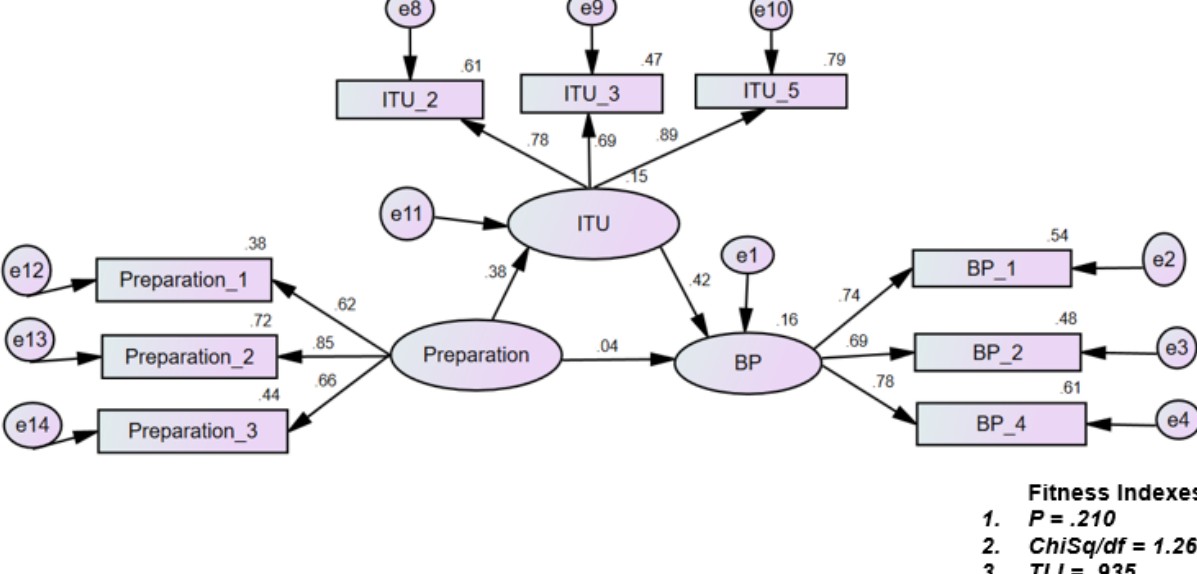

**Figure 4.** Representation of mediation analysis (Preparation -> ITU -> BP). Note: All fitness indexes achieved the required level. Source: Figure 4 generated through SPSS Amos 24.

**Table 8.** Mediating analysis summary (Preparation -> ITU -> BP).

| Relationship | Direct Effect | Indirect Effect | Confidence Interval | | *p*-Value | Conclusion |
| --- | --- | --- | --- | --- | --- | --- |
| | | | Lower Bound | Upper Bound | | |
| Awareness -> Intention to Use (ITU) -> Business Performance | 0.041 (0.808) | 0.269 | 0.018 | 1.165 | 0.036 | Full Mediation |
| | | Model Fit | | | | |
| CMIN/df = 1.268, GFI = 0.952, CFI = 0.956, TLI = 0.935, SRMR = 0.045 and RMSEA = 0.032. | | | | | | |

### 7.5. Serial Mediation Analysis

The researchers applied serial mediation to determine the impact of awareness (1) and preparation (2) on business performance through intention to use and government support, and later intention to use and skills as serial mediators.

Figure 5 and Table 9 represent the serial mediation analysis (Awareness -> ITU -> Skills -> BP). The study assessed the serial mediating role of intention to use and skills on the relationship between awareness and business performance. The result revealed an indirect effect of awareness on business performance through intention to use and skills. The result shows a positive and significant impact (b = 0.052, t = 2.821, *p* = 0.043), supporting H$_6$. However, in the presence of a mediator, awareness and business performance's direct effects were found to be insignificant (b = 0.300, *p* = 0.055). This suggests that intention to use and skills fully mediate the relationship between awareness and business performance.

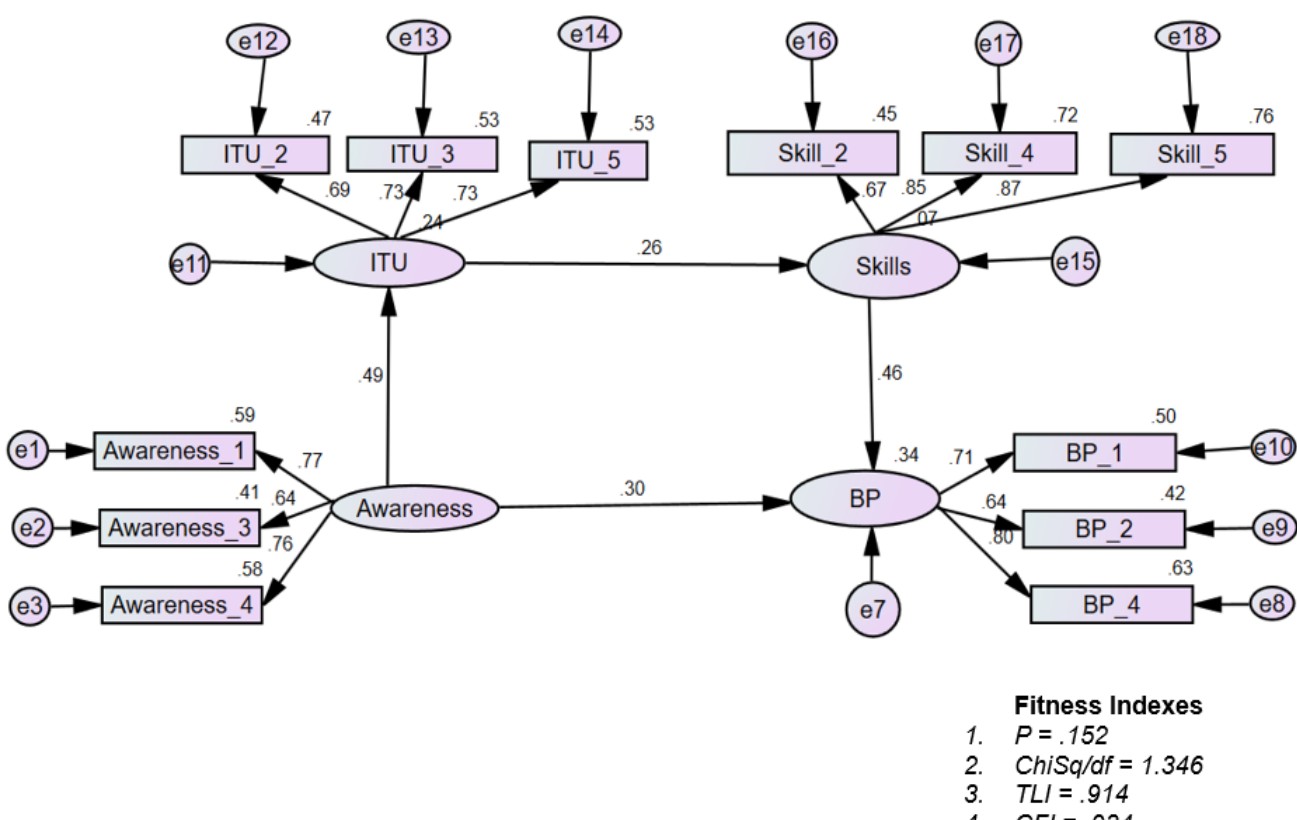

**Figure 5.** Representation of serial mediation analysis (Awareness -> ITU -> Skills -> BP). Note: All fitness indexes achieved the required level. Source: Figure 5 generated through SPSS AMOS 24.

**Table 9.** Serial mediation analysis summary (Awareness -> ITU -> Skills -> BP).

| Relationship | Direct Effect | Indirect Effect | Confidence Interval Lower Bound | Upper Bound | *p*-Value | Conclusion |
|---|---|---|---|---|---|---|
| Awareness -> Intention to Use (ITU) -> Skills -> Business Performance | 0.300 (0.055) | 0.052 | 0.002 | 0.164 | 0.043 | Full Mediation |
| Model Fit | | | | | | |
| CMIN/df = 1.346 GFI = 0.901, CFI = 0.934, TLI = 0.914, SRMR = 0.062 and RMSEA = 0.045. | | | | | | |

Figure 6 and Table 10 represent the serial mediation analysis (Awareness -> ITU -> GS -> BP). The study assessed the serial mediating role of intention to use and government support on the relationship between awareness and business performance. The result indicates an indirect effect of awareness and business performance through intention to use and government support, showing a positive and significant impact (b = 0.042, t = 2.393, *p* = 0.045), supporting H$_7$. However, the study found insignificant direct effects

on awareness and business performance in the presence of mediators (b = 0.285, *p* = 0.071). This suggests that intention to use and government support fully mediate the relationship between awareness and business performance.

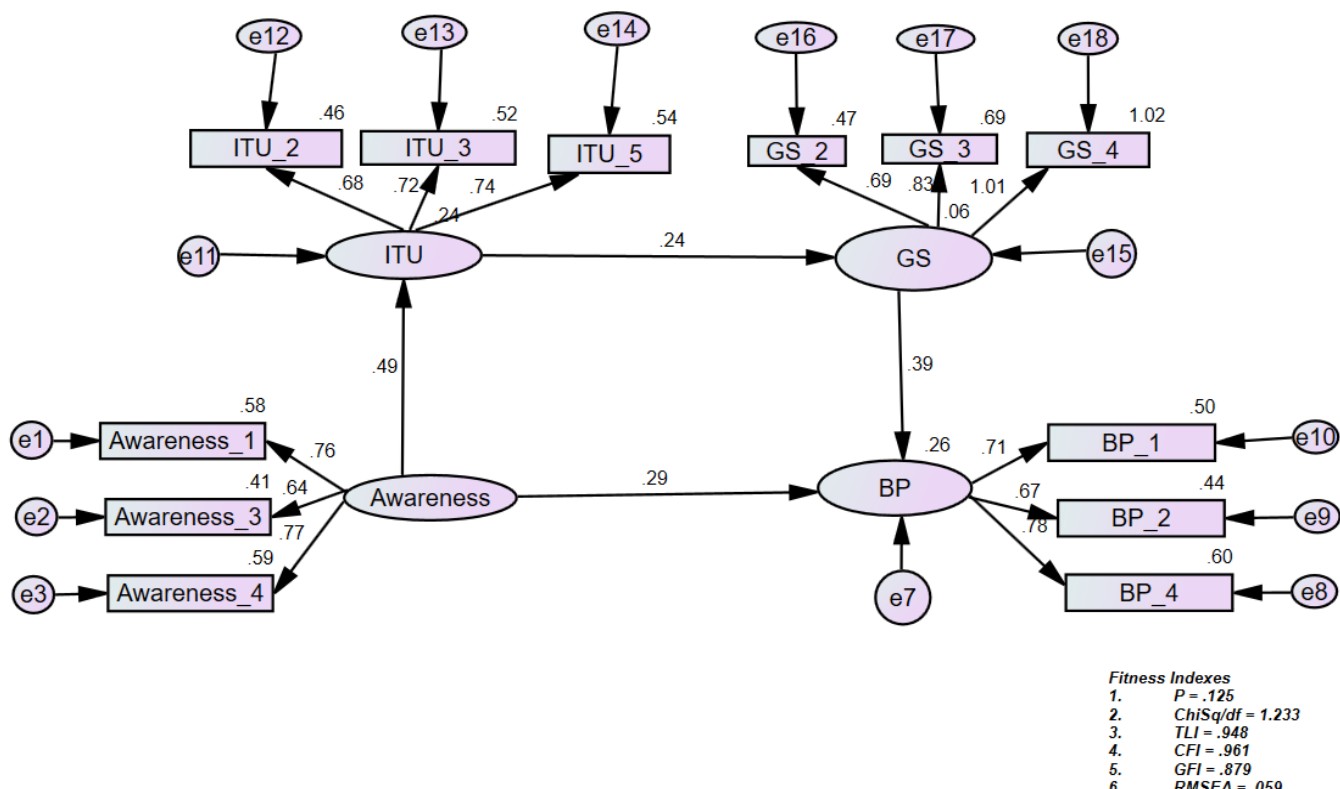

**Figure 6.** Representation of serial mediation analysis (Awareness -> ITU -> GS -> BP). Note: All fitness indexes achieved the required level. Source: Figure 6 generated through SPSS AMOS 24.

**Table 10.** Serial mediation analysis summary (Awareness -> ITU -> GS -> BP).

| Relationship | Direct Effect | Indirect Effect | Confidence Interval | | *p*-Value | Conclusion |
|---|---|---|---|---|---|---|
| | | | Lower Bound | Upper Bound | | |
| Awareness -> Intention to Use (ITU) -> Government Support (GS) Business Performance (BP) | 0.285 (0.071) | 0.042 | 0.001 | 0.133 | 0.045 | Full Mediation |
| Model Fit: CMIN/df = 1.233 GFI = 0.879, CFI = 0.961, TLI = 0.948, SRMR = 0.072 and RMSEA = 0.059 | | | | | | |

Figure 7 and Table 11 represent the serial mediation analysis (Preparation -> ITU -> Skills -> BP). The study assessed the serial mediating role of intention to use and skills on the relationship between preparation and business performance. The result reveals an indirect effect of preparation and business performance through intention to use, and skills were found positive and significant impact (b = 0.132, t = 2.567, *p* = 0.021), supporting $H_8$. The study found insignificant direct effects of awareness and business performance in the presence of a mediator (b = 0.074, *p* = 0.629). This suggests that intention to use and skills fully mediate the relationship between preparation and business performance.

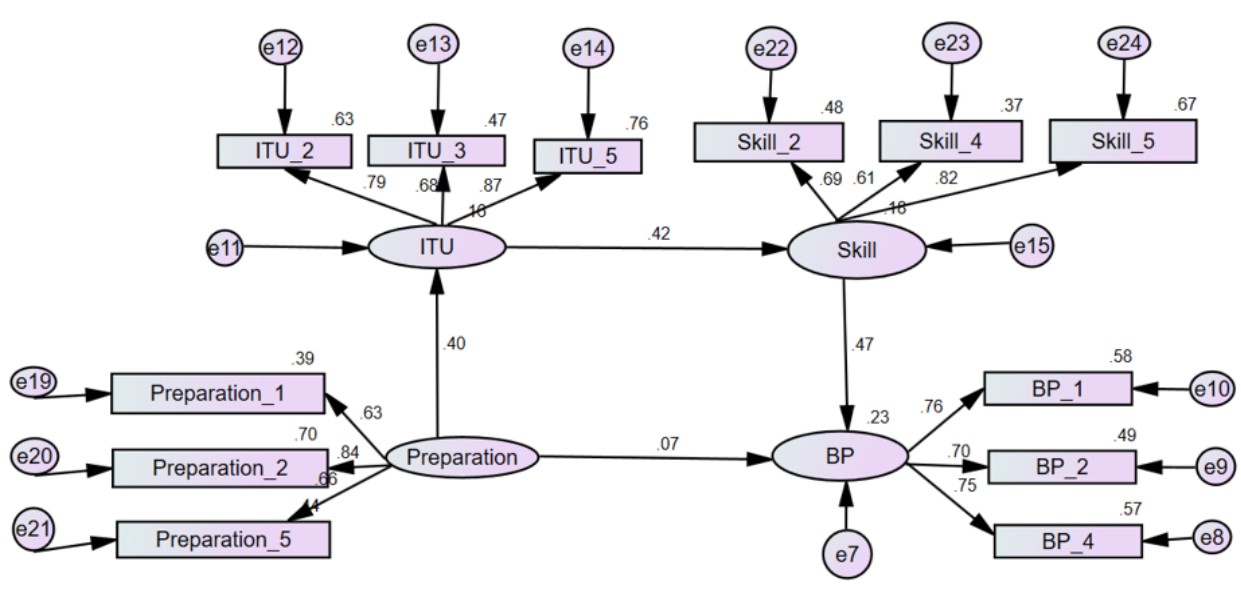

**Figure 7.** Representation of serial mediation analysis (Preparation -> ITU -> Skills -> BP). Note: All fitness indexes achieved the required level. Source: Figure 7 generated through SPSS AMOS 24.

**Table 11.** Serial mediation analysis summary (Preparation -> ITU -> Skills -> BP).

| Relationship | Direct Effect | Indirect Effect | Confidence Interval | | *p*-Value | Conclusion |
| --- | --- | --- | --- | --- | --- | --- |
| | | | Lower Bound | Upper Bound | | |
| Preparation -> Intention to Use (ITU) -> Skills -> Business Performance | 0.074 (0.629) | 0.132 | 0.013 | 0.509 | 0.021 | Full Mediation |
| Model Fit | | | | | | |
| CMIN/df = 1.282 GFI = 0.921, CFI = 0.944, TLI = 0.927, SRMR = 0.072 and RMSEA = 0.057. | | | | | | |

Figure 8 and Table 12 represent the serial mediation analysis (Preparation -> ITU -> GS -> BP). The study assessed the serial mediating role of intention to use and government support on the relationship between preparation and business performance. The result reveals a significant indirect effect of preparation and business performance through intention to use and government support, which were found positive and significant impact (b = 0.082, t = 2.224, *p* = 0.024), supporting H$_9$. The study found an insignificant direct effect on preparation and business performance in the presence of mediators (b = 0.169, *p* = 0.453). It suggests that intention to use and government support fully mediate the relationship between preparation and business performance.

Table 13 represents the summary of all the mediators and serial mediators. The Table results show that full individual mediation (intention to use) and full serial mediation (intention to use and government support, and intention to use and skills) exist in the study. The study also reveals that intention to use, government support, and skills are important mediators that help SMEs in successful digital transformation with awareness as well as preparation.

**Table 12.** Serial mediation analysis summary (Preparation -> ITU -> GS -> BP).

| Relationship | Direct Effect | Indirect Effect | Confidence Interval | | *p*-Value | Conclusion |
|---|---|---|---|---|---|---|
| | | | Lower Bound | Upper Bound | | |
| Preparation -> Intention to Use (ITU) -> Government Support (GS) -> Business Performance (BP) | 0.169 (0.453) | 0.082 | 0.009 | 0.475 | 0.024 | Full Mediation |

Model Fit
CMIN/df = 1.140 GFI = 0.891, CFI = 0.976, TLI = 0.968, SRMR = 0.074 and RMSEA = 0.046

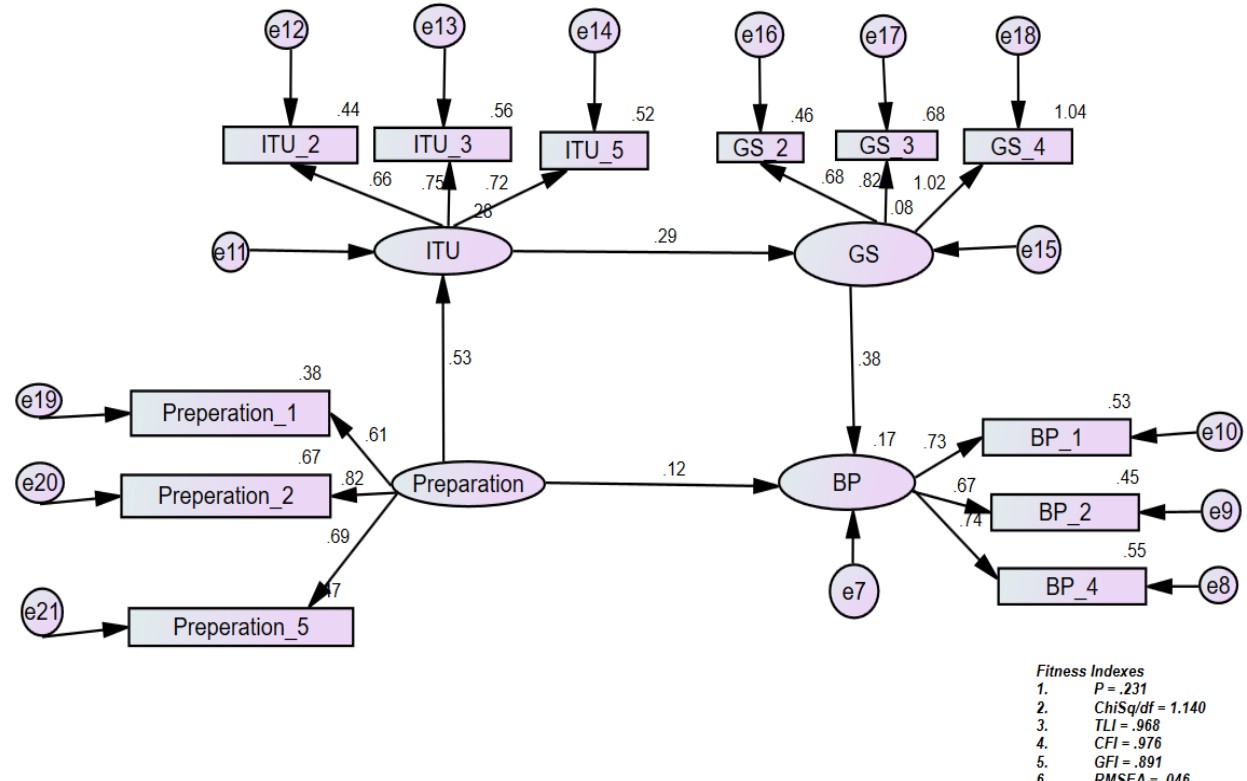

**Figure 8.** Representation of serial mediation analysis (Preparation -> ITU -> GS-> BP). Note: All fitness indexes achieved the required level. Source: Figure 8 generated through SPSS AMOS 24.

**Table 13.** Hypothesis result summary.

| Mediation Analysis Summary Result | | | | |
|---|---|---|---|---|
| Hypothesis | Indirect Effect | *p*-Value | Results | |
| H4: Awareness -> ITU -> BP | 0.151 | 0.027 | Supported | Full mediation |
| H5: Preparation -> ITU -> BP | 0.269 | 0.036 | Supported | Full mediation |
| H6: Awareness -> ITU -> Skills -> BP | 0.002 | 0.043 | Supported | Full serial mediation |
| H7: Awareness -> ITU -> GS -> BP | 0.042 | 0.045 | Supported | Full serial mediation |
| H8: Preparation -> ITU -> Skills -> BP | 0.132 | 0.021 | Supported | Full serial mediation |
| H9: Preparation -> ITU -> GS -> BP | 0.082 | 0.024 | Supported | Full serial mediation |

## 8. Discussion

SMEs are essential to economies because they significantly increase employment, innovation, and economic growth, and have been game-changers; using digital technologies helps them boost their competitiveness and sustainable economic development [116]. The

research emphasizes how SMEs' awareness and preparation for digitization impact their business performance, which helps them to create economic sustainability in the country. Some SMEs know that digitization suits their business and helps them grow [117]. Its advantages are well understood, but to make it successful, they need more skills, time, and financial resources [74]. Therefore, digital tools must be used appropriately and effectively to increase awareness [118]. Previous researchers observed that managers often must be made aware of the many digital transformation options and elements they must consider [50,119]. The study results show a significant relationship between awareness and intention to use (supported H1, Figure 2), which shows that SMEs' awareness towards digital transformation significantly and positively impacts intention to use following TRA and TPB [102,104]. SME performance is affected substantially by SME preparation for digital transformation aligning with TAM [101] (Supported $H_2$, Figure 2). Various empirical studies have shown that technology can improve productivity, innovation, international market access, and operational efficiency [92,101–103,120,121]. This study's results show that intention to use (digital technology) significantly impacts business performance (Supported $H_3$, Figure 2), and the results are consistent with the previous studies, which show a positive relationship between entrepreneurship intention and business growth following TAM [102,122,123].

Digitization benefits SMEs through cost reduction and improved production capacity [50,75]. By embracing digital transformation, SMEs can improve business results, productivity, and output [75]. Therefore, companies must design an innovative business model that enhances their business survival and ecological adaptability to ever-changing social norms, technological advancements, and product market conditions, as per TPB [102,124]. The study also supports $H_4$ and reveals that the intention to use fully mediates the relationship between awareness and business performance (Figure 3, Table 7). Furthermore, the researcher found that ($H_5$) intention to use fully mediates the relationship between preparation and business performance (Figure 4, Table 8). It shows that intention to use is one of the most valuable factors that helps improve business performance. To get a more in-depth study, the researcher applied serial mediation to know how skills and government support the intention to use impact business growth. The results show full mediation between Awareness -> ITU -> Skills -> BP (supported $H_6$, Figure 5, Table 9); Awareness -> ITU -> GS -> BP (supported $H_7$, Figure 6, Table 10); Preparation -> ITU -> Skills -> BP (supported $H_8$, Figure 7, Table 11); and Preparation -> ITU -> GS -> BP (supported $H_9$, Figure 8, Table 12). The study emphasizes how digital transformation affects SMEs in Saudi Arabia, particularly concerning adopting new technologies, value creation, and developing unique skills that improve the market position and significantly benefit the company's growth [94].

The study results show that SMEs in Saudi Arabia will benefit from utilizing and adopting digital technologies and their benefits and potential uses [102]. Furthermore, the study has scope for SMEs to explore the role of government and skills in managing the firm's performance. Digitization is the key to survival as SMEs adapt to the global economy and take initiatives to launch specific training programs and incentives for SMEs to boost digitization [101]. The Small and Medium Enterprises General Authority (Monsha'at) collaborates with various academic institutions to train SMEs in their region. Implementing training initiatives and projects could create a win-win situation for the SMEs in Saudi Arabia. The government could also conduct hackathon events among SMEs to create digital awareness. It is essential to support SMEs' growth and help small businesses gain a competitive advantage through special programs, funding, policies, and counseling on account of their planned behavior [101,125] and strengthen their support for digital transformation goals through technology acceptance [102,103,126]. Fachrunnisa et al. (2013) [98] found that SMEs perceive more incredible benefits or outcomes when collaborating virtually. The government and other stakeholders must support this community network in several ways. The World Economic Forum, in collaboration with the Ministry of Communication and Information Technology Kingdom of Saudi Arabia [99], published a paper on modernizing

SMEs in 2023, mentioning five National Industrial Strategy (NIS) initiatives to support digital technology adoption in the SMEs' advanced manufacturing industry, the data also show that most people in the kingdom have basic ICT skills. Our study found that most SMEs have basic ICT skills and are aware of new digital tools, but there is a need to focus on implementing these tools. In addition, some SMEs are familiar with the significance of SME digitalization, but need more practice to implement it. Therefore, it is essential to work on building those skills and implementing digitalization practices in these SMEs in the country. The ministry could also tie up various PPPs (Public–Private-Partnerships) in the region to create and improve digital skills. It would be better to provide two types of consultancies to the SMEs; one is a management consultancy, which helps them to design relevant strategies, and the other one is a technical consultant, who specializes in the digital section. The digital consultants would help smoothen the SME's digital journey, and SME decision-making capacities driven by data are being strengthened by digital technology, enabling them to innovate and grow their enterprises in ways that promote sustainable economic and human development [116]. SMEs' businesses flourish due to the digitization process; it improves the performance of small and medium-sized businesses [117]. However, they require more skills, time, and resources to implement successfully.

Previous data and reports show that sustainable economic development is primarily concerned with ensuring that people with low incomes have access to secure and sustainable livelihoods at the national level; this calls for policies to encourage environmentally responsible economic conduct [127]. Digitalization's prospects for creating a sustainable society of the future are outlined in this perspective, as stated by Maria E. Mondejar et al. (2021) [33]. One aspect of society's overall development is its economic development, which is linked to both institutional and technological (digital) transformation as well as economic growth [127]. A fair, ecologically sustainable, and healthy society can be ensured by proactively addressing difficulties related to the Sustainable Development Goals (SDGs) of the United Nations through the development of smart systems linked to the Internet of Things. When it comes to securing sustainable economic growth, digital sustainability refers to the endeavor of creating and implementing smart technologies. It is predicted that by 2030, modern digital technologies will contribute 14% to the global economy [128,129]. A number of advantages are also provided by combining dynamic capabilities with technologies, such as increased market competitiveness, better operational effectiveness, cost advantage, high market share, reduced wastage, knowledge data, and innovation.

## 9. Practical Implications

The study has implications for Saudi authorities to improve the existing design of digital transformation guidelines for SMEs, which would guide them in successfully implementing digitization. The authorities can also start a mentoring program where the ministries appoint consultants to support and sustain the SME's digital transition. Such support will boost the SMEs' morale and be a silver lining. The authorities can also arrange numerous consultancy programs at a subsidized rate for the expansion and development of SMEs. The study also has implications for existing companies to start their digital journey by implementing digital platforms using social media platforms at a low cost.

## 10. Conclusions

To recapitulate, the mediation results of the study depict that skills and government support are crucial for implementing digital tools to lead to business growth. In addition to awareness and preparation, intention to use plays a prominent mediating role in achieving a desirable SME performance with favorable government support and required skills. As is evident from the study results, SMEs should understand the importance of digitalization and its implementation within their enterprises. Through digitalization, SMEs can take advantage of competitive advantages, promote teamwork, and succeed over the long run in their sustainability-driven digital transformation process. SMEs can access a range of viewpoints, expertise, and experiences by incorporating stakeholders in the innovation

process, leading to the development of more customized and pertinent solutions. SMEs can increase their market reach, improve operational efficiency, and match their strategy to sustainable objectives by leveraging these capabilities, positioning themselves for success in their quest for sustainability, and utilizing digital capabilities. The novelty in the research is the inclusion of two primary mediators, i.e., government support, skills, and intention to use, which shows and proves that increased support from the government improves skills, which leads to improved business performance.

Moreover, the study has a significant outcome for the authorities to understand how skills and government support are the vital factors that must be prioritized for the successful implementation of the National Transformation Program, which contributes to achieving Saudi Vision 2030, and guide them accordingly.

## 11. Limitations and Prospects for Future Research

Firstly, the study is confined to 68 Saudi Arabian SMEs in the Ha'il region. In addition, the data were collected for a certain period due to a restriction on project duration. An extended study on digitization could be conducted by comparing GCC countries. Secondly, the study uses a closed-ended questionnaire to understand SMEs' awareness and preparation. SMEs' awareness, preparation, and perceptions regarding digitization can be explored further using an open-ended questionnaire or in-depth or focus group interviews. The study could also be extended using SMEs' barriers to get more insight into it. Thirdly, the study could be extended for further detailed studies on gender diversity, providing more insight into successfully implementing digitization. Researchers could also have scope to explore women's entrepreneurship and empowerment in small and medium enterprises on digital technologies and their successful implementation.

**Author Contributions:** Conceptualization, A.T. and A.S.; methodology, A.T.; software, A.T.; validation A.T.; formal analysis, A.T.; investigation, A.T. and A.S.; resources, A.T. and A.S; data curation, A.T. and A.S.; writing—original draft preparation, A.T.; writing—review and editing, A.T. and A.S.; project administration, A.T. and A.S.; funding acquisition, A.T. and A.S. All authors have read and agreed to the published version of the manuscript.

**Funding:** This paper is a part of the approved research project funded by Dr. Nasser Al-Rasheed Chair for Future Pioneers, the Scientific Research Chairs at the University of Ha'il, Saudi Arabia (Project No. SCR-22 061), titled "SMEs Awareness and Preparation for Digital Transformation: Exploring Business Opportunities for entrepreneurs in Saudi Arabia's Ha'il Region". The authors would like to express their gratitude to the Scientific Research Chairs of the University of Ha'il, Saudi Arabia, for funding this study.

**Institutional Review Board Statement:** The study was conducted in accordance with the Declaration of Helsinki, and approved by the Institutional Review Board (or Ethics Committee) of the University of Ha'il, kingdom of Saudi Arabia (protocol code H-2023-427 and dated 4/12/2023).

**Informed Consent Statement:** Informed consent was obtained from all subjects involved in the study. The researchers distributed an informed consent form regarding the survey to the SME respondents. Respondents ensured that no data or information they provided would be disclosed in the research without their consent.

**Data Availability Statement:** The data will be made available on request from the corresponding author.

**Conflicts of Interest:** The authors declare no conflicts of interest.

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
