# Peer review of "SMEs Awareness and Preparation for Digital Transformation: Exploring Business Opportunities for Entrepreneurs in Saudi Arabia’s Ha’il Region"

_sustainability, doi:10.3390/su16093831_

Round 1
Reviewer 1 Report
Comments and Suggestions for Authors
This submission attempts to explore a local region digital transformation readiness and related effects on their business operations. Major observation:
--+ Abstract provides some information about the study, especially the assessment and evaluation results, but is very poor to convey the rationale and key implementation details for superior research outcomes.
+- Introduction is good to establish a conceptual background, with relevant references, etc. but fails to present a comprehensive view of this study, instead of anticipated benefits of the ongoing efforts. Also, some repetitious statements are suggested to be revised for a smooth flow (seems to be the case for the whole submission to some extent).
-- Literature review touches on many relevant concepts with references. However, some references or subsections were summarized plainly in a shallow manner, discreetly, or uncohesively, without offering insightful inter/related perspective(s) or without closing/integrating the loop, e.g., lines 123-126. Also, this section completely misses comparative studies (with a sentence on line 218 only).
-- Table 1 presentation is extremely poor and needs to be revised significantly to improve its efficacy and relations to the current activity. Perhaps, an additional diagram could be developed to clarify inputs, tools, outputs, etc., in addition to the content&format revised Table 1. Also, additional information regarding to the component selection justifications, enumeration of the components, individual used component justifications/details/relevance, etc. should be presented to understand the tools and their merits, if any.
+- Various theories that are used in this study are briefly summarized. But, related theoretical results from other studies, accuracy levels, potential strengths and weaknesses of the utilized theories, etc. need to be presented so that the merits of the tools can be evaluated.
- Sections 4-5 need some narrative to guide the reader.
- Three survey questionnaire reviewers need to be described further to see their background, relevance, etc., for the current study.
- The measurement model in Fig. 1 is not effective to describe the underlying process. The number of (68) survey respondents need to be compared with the total number so that statistically significant results can be derived. As the goal is to understand the digitalization, the Google survey may not reach out the real participants who hesitate the technology usage. Sampling of the respondents and assessments need to consider these important details for illustrative data. Also, "using Judgmental non-probability sampling " should be explained to see its impact on the activity. The respondent "Experience in entrepreneurship" profile does not seem to indicate a uniform distribution to obtain reliable/meaningful results.
- The 22-item questionnaire need to be listed in the draft so that each question relevance can be studied.
- The statistical structural model in Fig. 2 needs to be legible and needs to be explained its creation by the software, components, etc., for a typical reader.
--+ The SPSS model results and related explanations seem to be following a pattern of software-generated structural model and respective analysis. The Discussions in section-8 seem to be scattered for the results and partially un/related to the current research topic., i.e., the research activity related results and associated discussions seem to be shifting into much broader discussions (not in the activity though).
-- The Conclusion section should be all-text only and should be wrapping up the current activity while the actual conclusion is a collection of many different (mostly unexpected) components.
--- The activity can potentially extract much useful info/guidance about the particular part of the particular country. However, the research motivation, focus, development, implementation, results, etc. seem to be missing/shifting/shuffling research-oriented focus for meaningful and strong results..
- Unexpected presentation issues: - Expand all acronyms in their first usage, - "Intension " (seems to be improperly used), - "frameworks pro-competition ", - "Due o the lack of funding ", - " Less than 2 (Ullman, 2001)[79] to 5 ", - "scholars. [57]. ".
- Unexpected presentation issues: - Expand all acronyms in their first usage, - "Intension " (seems to be improperly used), - "frameworks pro-competition ", - "Due o the lack of funding ", - " Less than 2 (Ullman, 2001)[79] to 5 ", - "scholars. [57]. ".
Author Response
Dear Reviewer,
Kindly find the point-to-point response with attached file.
Thank you.
Reviewer 2 Report
Comments and Suggestions for Authors
Thank you for the opportunity to review this paper on awareness of digitalisation and intention to digitise SMEs. The digitalisation topic is very trendy and could pose interesting point of view. However, the novelty of the article is questionable. As indicated in the notes below, the article needs a thorough revision to strengthen the novelty and validity of empirical research.
Notes on specific details:
Sustainability appears only in the beginning and at the end of the article. Reading sustainability related paragraphs seem to be artificially integrated into the text, adding no scientific value to the topic.
I feel that the language style lacks clarity. I.e. use of awareness and intention to use without the subject of digitalisation throughout the text may raise questions what is being discussed. The "gap" is presented unclearly: is it a gap only in Saudi Arabia or globally? If only in Saudi Arabia the analysed connections of the concepts are misrepresented, clearer indications of international studies are missing.
The validity of empirical study is questionable, as only 68 respondents were included into the survey. Such sample would suit pilot study better. More detailed sample description is also lacking. The study focuses on organisations, yet the sample is detailed for people, not organisational statistics. Is "Intention to use" truly regarded as mediator in the measurement model (fig. 1)? The items are not described in detail. The questionnaire is not provided. The results are presented in a hectic manner, making it harder to double check information for validity assurance: i.e. it is indicated in the notes of fig. 8 that "All fitness indexes achieved the required level", however, GFI is less than 0,9.
As the moderation/mediation effect is mixed up, I am unable to determine the soundness of the results and discussion.
Technicalities:
Line 39 misses punctuation.
2.2. does not define or explain Intention to Use.
Line 220 which researcher?
The name of the country should be used in full.
Citing in text must be improved following the citing recommendations and citing style of the journal.
Chapters 3, 4 and 5 should be incorporated into chapter 2.
If abbreviations are introduced to text, they should be used without the full phrase.
Comments on the Quality of English LanguageEnglish is used adequately. The overall style of the text lacks clearer structure.
Author Response

(The authors gave the same response as above.)

Reviewer 3 Report
Comments and Suggestions for Authors
The subject of the article is certainly interesting. However, it is not possible to assess whether the purpose of the study as stated in the introduction of the paper has been achieved. This is because the author does not explain what the constructs Awareness_1..., Preparation_1...., BP_1..., ITU_2 measure, what questions were asked to the respondents, whether the sample of 68 respondents is representative of the total number of SMEs in Saudi Arabia. In this regard, the scheme in the 6.4 Measurement Model and others resulting from is difficult to understand.
Additional comments and suggestions:
1. The abstract of the article should be rewritten: please remove redundant abbreviations, add the information about the relevance of the topic, state the general purpose, briefly describe hypothesis and the results obtained.
2. Please find a more appropriate place for sections 4 and 5 by showing the connection between the theoretical background and the hypotheses derived by the author.
Author Response

(The authors gave the same response as above.)

Reviewer 4 Report
Comments and Suggestions for Authors
The study entitled “SMEs Awareness and Preparation for Digital Transformation: Exploring Business Opportunities for Entrepreneurs in Saudi Arabia’s Ha’il Region” has claimed that study has been reviewed and approved by the Research Ethics Committee at University of Hail dated on 4th December 2023. The study has highlighted the contemporary issue in the field. However, there are some concerned about this study.
1. There is no need to mention the no. of items in the abstract.
2. There are grammatical mistakes in the study e.g. in line 39 author has mentioned 46 per cent.
3. Add some facts about the importance of study in the introduction.
4. Write one paragraph about the structure of study at the end of introduction.
5. Make research questions Instead of writing the objectives
6. Kindly bold the Table and Figure numbers in the study to highlight to attract the attention of reader.
7. Do not copy paste the pictures from AMOS eg. Figure 2. Improve the diagrams
8. In Table 6, last rows values must be write below the table e.g. CMIN to RMSEA like mentioned in Table 7. But all the tables and figures needs proper formatting.
9. Support the study with some recent studies.
Comments on the Quality of English LanguageNeeds to improve
Author Response

(The authors gave the same response as above.)

Reviewer 5 Report
Comments and Suggestions for Authors
This study is innovative. However, there are still several problems as follows:
1, the importance of digital transformation of SMEs is argued at the beginning of the literature review, but the relationship between digitalization and sustainability is argued weakly.
2, the study says in the abstract that 68 SMEs were surveyed, but in the sample section, it says there were 68 respondents. Does this mean that there is only one representative from a business and some have only been working for a year? It is hoped that these figures can be further supplemented.
3. hopefully it can be added that there is sufficient justification before hypotheses are made.
4. I would like to encapsulate all the assumptions in one figure. Figure 1 does not correctly represent H4 and H5.
5. Figure 2. is shaded on both sides.
Author Response

(The authors gave the same response as above.)

Round 2
Reviewer 1 Report
Comments and Suggestions for Authors
The revised manuscript seems to contain minor improvements while major concerns seem to remain. Literature review is necessary but not sufficient to enhance the results in this study. Some literature descriptions are not effectively supporting the current activity methodology, e.g., 2.1, 2.2, and 2.3 should explore specific linkages to the current study in terms of questions, methodologies, analysis, etc., for viable results for the particular region, especially the ones stated in the limitations section of the current activity, e.g., how did they setup their respective studies?, how did those studies find most accurate components/relationships for all segments of the society?, what tools/approaches/assessments did they utilize to obtain the most relevant/viable information?, etc.. In their current forms, these sections establish research results only and do not guide the current activity for superior results.. Also, lack of survey effectiveness details, concerns about statistical significance of the results, etc. raise concerns for comprehensive outcomes for this particular region digital transformation awareness and preparations.
Comments on the Quality of English LanguageMinor corrections such as acronyms, etc. should be addressed.
Author Response
Dear reviewer,
We have attached the responses to comments in attached MS word file.
Please check the file.

Reviewer 2 Report
Comments and Suggestions for Authors
I appreciate the authors' efforts to address some of my concerns in the revised manuscript. However, several major issues remain that require further attention before I can recommend publication in Sustainability.
Main Concerns:
1. The justification for presenting this work in the context of "sustainable economic growth" remains unconvincing. "Sustainable economic growth" requires resource decoupling, which is absent from the discussion. The authors must either clearly define their use of "sustainability" or demonstrate how their work contributes to resource decoupling and sustainable development.
2. The selection and description of the SME sample lack factual support. The authors do not demonstrate that these SMEs represent the broader population of SMEs in the region or country.
3. The logic behind the organization of theories, research questions, and hypotheses needs clarification. Research questions should be formulated as actual questions, not statements.
4. The description of the research methods contains several inconsistencies. The number of questionnaire items in the text (22) differs from the number presented in Table 1.
5. Abbreviations are still used inconsistently throughout the manuscript. Either consistently define and use abbreviations or avoid them altogether.
6. Although a table of responses to reviewer comments is provided, not all suggested revisions are implemented. Authors should clearly indicate which lines of the manuscript were amended and ensure all feedback is addressed.
Additional Comments:
· The manuscript would benefit from improved citation practices.
· The intention to use deserves more than a single sentence to warrant inclusion in the subchapter title.
Recommendation:
The manuscript requires significant revisions to address the concerns raised above before it can be considered for publication in Sustainability. I encourage the authors to carefully review my feedbacks and provide a revised manuscript that demonstrates a substantial improvement in these areas.
Comments on the Quality of English LanguageI still find misuse of abbreviations and differently used country name throughout the text.
Additionally, style of English could be improved to bring more clarity when bringing the message.
Author Response
Dear Reviewer,
We have attached the responses comments in attached MS word file.
Please check the file.

Reviewer 3 Report
Comments and Suggestions for Authors
The author has responded and provided an explanation to my comments. It has mostly met my requirements.
Author Response
Dear Reviewer,
As you already mentioned that the author has responded and provided an explanation to my comments. It has mostly met my requirements.
Thanks.
Reviewer 5 Report
Comments and Suggestions for Authors
The authors have successfully addressed all my concerns. I thus recommend acceptance of this paper.
Author Response
Dear Reviewer,
You already mentioned that the authors have successfully addressed all my concerns. I thus recommend acceptance of this paper.
Many thanks.
Round 3
Reviewer 1 Report
Comments and Suggestions for Authors
The revised draft contains some improvements, e.g., enhancements in the narrative and research questions, while the author response is very ineffective to address the concerns about research question creations, effectively linking to existing studies for superior outcomes in many aspects (could be fixed but rationale developments are essential), related 22-survey question developments (seems to be missing after Table-1 to link to the research questions), survey population selection (the stated subjective treatment is very much troubling), etc.
Comments on the Quality of English LanguageMinor adjustments are needed.
Author Response
Dear Reviewer,
Please find the author responses for comments in the attached file.
Regards,
